

# *In situ* measurements of perturbations to stratospheric aerosol and modeled ozone and radiative impacts following the 2021 La Soufrière eruption

Yaowei Li[1,#], Corey Pedersen[1,#], John Dykema[1], Jean-Paul Vernier[2,3], Sandro Vattioni[4], Amit Kumar
Pandit[3], Andrea Stenke[4,5,6], Elizabeth Asher[7,a], Troy Thornberry[8], Michael A. Todt[7,b], ThaoPaul Bui[9],
Jonathan Dean-Day[10], Frank N. Keutsch[1,11,12]

[1]School of Engineering and Applied Sciences, Harvard University, Cambridge, Massachusetts 02138, United States
[2]NASA Langley Research Center, Hampton, Virginia 23666, United States
[3]National Institute of Aerospace, Hampton, Virginia 23666, United States
[4]Institute of Atmospheric and Climate Science, ETH Zürich, 8092 Zurich, Switzerland
[5]Institute of Biogeochemistry and Pollutant Dynamics, ETH Zurich, 8092 Zurich, Switzerland
[6]Eawag, Swiss Federal Institute of Aquatic Science and Technology, 8600 Dübendorf, Switzerland
[7]Cooperative Institute for Research in Environmental Sciences (CIRES), University of Colorado, Boulder, Colorado 80309,
United States
[8]Chemical Sciences Laboratory, National Oceanic and Atmospheric Administration (NOAA), Boulder, Colorado 80309,
United States
[9]NASA Ames Research Center, Moffett Field, California 94043, United States
[10]Bay Area Environmental Research Institute, Petaluma, California 94035, United States
[11]Department of Chemistry and Chemical Biology, Harvard University, Cambridge, Massachusetts 02138, United States
[12]Department of Earth and Planetary Sciences, Harvard University, Cambridge, Massachusetts 02138, United States
a Now at Global Monitoring Laboratory, National Oceanic and Atmospheric Administration (NOAA), Boulder, Colorado
80309, United States
b Now at Finnish Meteorological Institute (FMI), 00560 Helsinki, Finland
[#]These authors contributed equally to this work

*Correspondence to*: Yaowei Li (yaoweili@seas.harvard.edu), Frank N. Keutsch (keutsch@seas.harvard.edu)

**Abstract.** Stratospheric aerosols play important roles in Earth's radiative budget and in heterogeneous chemistry. Volcanic eruptions modulate the stratospheric aerosol layer by injecting particles and particle precursors like sulfur dioxide ($SO_2$) into the stratosphere. Beginning on April 9[th], 2021, La Soufrière erupted injecting $SO_2$ into the tropical upper troposphere and lower
stratosphere, yielding a peak $SO_2$ loading of 0.3-0.4 Tg. The resulting volcanic aerosol plumes dispersed predominately over the northern hemisphere (NH), as indicated by the CALIOP/CALIPSO satellite observations and model simulations. From June to August 2021 and May to July 2022, the NASA ER-2 high-altitude aircraft extensively sampled the stratospheric aerosol layer over the continental United States during the Dynamics and Chemistry of the Summer Stratosphere (DCOTSS) mission. These *in situ* aerosol measurements provide detailed insights into the number concentration, size distribution, and
spatiotemporal variations of particles within volcanic plumes. Notably, aerosol surface area density and number density in 2021 were enhanced by a factor of 2-4 between 380-500 K potential temperature compared to the 2022 DCOTSS observations, which were minimally influenced by volcanic activity. Within the volcanic plume, the observed aerosol number density



exhibited significant meridional and zonal variations while the mode and shape of aerosol size distributions did not vary. The La Soufrière eruption led to an increase in the number concentration of small particles (<400 nm), resulting in a smaller aerosol

effective diameter during the summer of 2021 compared to the baseline conditions in the summer of 2022, as observed in regular ER-2 profiles over Salina, Kansas. A similar reduction in aerosol effective diameter was not observed in ER-2 profiles over Palmdale, California, possibly due to the already smaller values in that region during the limited sampling period in 2022. The La Soufrière eruption was modeled with the SOCOL-AERv2 aerosol-chemistry-climate model. The modeled aerosol enhancement aligned well with DCOTSS observations, although the direct comparison was complicated by issues related to

the model's background aerosol burden. This study indicates that the La Soufrière eruption contributed at most 0.6% to Arctic and Antarctic ozone column depletion in both 2021 and 2022, which is well within the range of natural variability. The modeled top-of-atmosphere one-year global average radiative forcing was -0.08 W/m$^2$ clear-sky and -0.04 W/m$^2$ all-sky. The radiative effects were concentrated in the tropics and NH midlatitudes and diminished to near-baseline levels after one year.

## 1 Introduction

Stratospheric aerosols are important to the climate and chemistry of the Earth (Deshler, 2008; Kremser et al., 2016 and references therein). They modify the global radiative budget by scattering and absorbing solar shortwave and terrestrial longwave radiation. Aerosols in the lower stratosphere also affect the cloud life cycle by acting as ice nuclei (Sporre et al., 2022). Studies have shown that approximately 21% of the overall direct aerosol radiative forcing since 1850 can be attributed to stratospheric aerosols (Yu et al., 2016). In general, stratospheric aerosols have a net surface cooling effect (Robock, 2000).

Stratospheric aerosols can also provide a surface for heterogeneous reactions to occur that amplify catalytic ozone loss cycles (Fahey et al., 1993; McNeill and Thornton, 2023; Solomon, 1999; Solomon et al., 2023). The climate and chemical impacts of stratospheric aerosols depend on their chemical composition, concentration, and size distribution (Li et al., 2021; Murphy et al., 2021).

Volcanic eruptions have a profound impact on the physicochemical properties of stratospheric aerosols by injecting a

substantial amount of ash and particle precursors, such as sulfur dioxide ($SO_2$), directly into the stratosphere (Friberg et al., 2018; Kremser et al., 2016; Schmidt et al., 2018). Once in the stratosphere, $SO_2$ undergoes oxidation within a span of weeks (Rex et al., 2014), resulting in the formation of sulfate aerosols (sulfuric acid and water mixture). These aerosols can persist for months to years in the stratosphere, depending on particle size and the location of the volcanic plumes, due to limited wet and dry deposition processes. Therefore, it is crucial to understand the variations in aerosol concentration and size distribution,

as well as the transport of the volcanic plumes after eruptions to accurately assess their impacts on climate and chemistry. Our current understanding of the size distribution of stratospheric volcanic aerosols is largely based on studies of Mt. Pinatubo eruption in 1991 which was the largest eruption (regarding $SO_2$) in the satellite era and resulted in a substantial increase in aerosol size (Deshler et al., 1993; Russell et al., 1996). However, the variability of stratospheric aerosol size distribution after volcanic eruptions, particularly those associated with moderate and small eruptions, remains unclear (Marshall et al., 2022).



These eruptions, despite their smaller size, have been shown to contribute to persistent variability in stratospheric aerosols (Vernier et al., 2011) and can have significant climate impacts (Andersson et al., 2015; Solomon et al., 2011).

Satellite remote sensing provides long-term, global coverage of stratosphere aerosol retrievals (Thomason et al., 2018). It has been the primary method for assessing the volcanic influence on stratospheric aerosols and provides critical inputs, such as stratospheric aerosol optical depth and extinction profiles, for models to evaluate the subsequent climate and chemical impacts

(Friberg et al., 2018). Satellite-derived stratospheric aerosol products, especially limb-scatter based measurements, rely on assumptions about the aerosol size distribution (e.g., Loughman et al., 2018). To ensure accurate satellite retrievals, *in situ* aerosol measurements are essential as they provide size-resolved stratospheric aerosol number concentrations. Balloon-borne measurements offer valuable datasets on aerosol size distribution but are limited in spatial coverage (e.g., Kalnajs and Deshler, 2022; Todt et al., 2023). On the other hand, aircraft measurements provide wider spatial coverage and, in certain cases, finer

temporal resolution, which are crucial for understanding the spatiotemporal variability of volcanic plumes and their impacts in the stratosphere (Pueschel et al., 1994; Wilson et al., 1993). While aircraft-based observations play a vital role in enhancing our precise understanding of volcanic plume evolution in the stratosphere, such observations are limited due to the challenge of organizing rapid and dedicated deployments following volcanic eruptions.

On April 9th, 2021, La Soufrière (13.3˚N, 61.2˚W) began erupting explosively (maximum volcanic explosivity index of four).

Horváth et al. (2022) identified 49 distinct eruptive events between April 9-22nd. The stratospherically relevant eruptive events occurred primarily between April 9-12th (Taylor et al., 2022). Peak $SO_2$ column loading was measured to be between 0.3-0.4 Tg (Bruckert et al., 2023; Taylor et al., 2022). The bulk of the plumes and $SO_2$ spread near the tropopause (~17-18 km) with overshooting tops reaching up to 23 km (Horváth et al., 2022; Taylor et al., 2022). Multi-angle Imaging SpectroRadiometer (MISR) observations also indicate that the volcanic ash plume reached levels above 20 km (NASA Earth Applied Sciences

Disasters Program).  The volcanic plumes were first detected over North America on April 30th by balloon-borne optical particle spectrometer launched from Boulder, Colorado (Todt et al., 2023).

Two months after the eruption of La Soufrière, the Dynamics and Chemistry of the Summer Stratosphere (DCOTSS) airborne mission obtained *in situ* measurements of aerosol number concentration and size in the lower stratosphere (13-21 km, 380-500 K potential temperature) over North America between June and August 2021. The extensive spatial and temporal coverage of

the DCOTSS dataset enables a detailed examination of the aerosol variation as volcanic plumes age and disperse in the stratosphere during the months following the eruption (2-4 months post-eruption) and will be useful for validating satellite observations and model simulations. Furthermore, DCOTSS observations between May and July 2022, a period when the volcanic influence was minimal, serve as a baseline state for comparing the volcanic perturbations observed in 2021.

In this study we analyzed the stratospheric aerosol perturbations from the 2021 La Soufrière eruption and impacts on

stratospheric ozone and radiative forcing, combining *in situ* aircraft and balloon-borne measurements, satellite observations, and SOCOL-AERv2 aerosol-chemistry-climate model simulations. We examined the distribution and transport of volcanic aerosol plumes in the stratosphere using satellite observations and model simulations. The modeled aerosol concentration enhancement was consistent with DCOTSS *in situ* aircraft measurements in northern hemisphere (NH) midlatitudes. Through



DCOTSS measurements and model simulations, we explored the spatial variations in aerosol number density and size
distribution within the volcanic plumes, as well as the impacts of these plumes on the stratospheric aerosol effective diameter.
We also used balloon-borne optical particle counter measurements to gain insights about aerosol perturbations above 20 km
in NH midlatitudes. Using our model simulations, we examined the radiative forcing and ozone column depletion caused by
the eruption.

## 2 Measurement and Model Descriptions

**2.1 DCOTSS Aircraft Measurements**

The DCOTSS campaign took place in June-August 2021 and May-July 2022 over North America, with a focus on
understanding the interactions between dynamical and chemical processes that determine the composition of the extratropical
summer stratosphere (https://dcotss.org/). With the NASA ER-2 high-altitude aircraft platform, aerosols and various trace
gases were sampled and analyzed *in situ* to characterize the background stratosphere and perturbations from overshooting
convective events, volcanic eruptions, and wildfire injections. Figure 1 shows the ER-2 flight tracks of 31 total flights during
the DCOTSS 2021 and 2022 missions. Most of the flights originated from Salina, Kansas (39° N, 98° W), while some flights
were conducted from Palmdale, California (35° N, 118° W). The ER-2 aircraft can reach altitudes as high as 22 km
(approximately 510 K potential temperature) and has a maximum flight duration of 8 hours. Vertical profiling between 13 and
22 km has been constantly conducted during the DCOTSS missions.

The DCOTSS Portable Optical Particle Spectrometer (DPOPS) instrument aboard the ER-2 aircraft measured the particle
number density and size distribution between 140 and 2,500 nm at 1Hz resolution (Li et al., 2022). The core part of the
instrument, i.e., the Portable Optical Particle Spectrometer (POPS, Handix Scientific, Boulder, CO), uses a 405 nm diode laser
to count and size individual particles (Gao et al., 2016). DPOPS is an optimized POPS system designed specifically for high-
altitude aircraft platforms, enabling isokinetic sampling of particles throughout the troposphere and lower stratosphere with
autonomous operation during flight. The DPOPS instrument was calibrated using size-classified dioctyl sebacate (DOS,
refractive index of $1.45+0i$) particles and polystyrene latex (PSL, refractive index of $1.615+0.001i$) beads before and after the
campaign, and the performance was routinely checked with 300 nm PSL beads after each flight. For this study, the size
distribution data calibrated with DOS was employed, as stratospheric volcanic aerosols predominantly consist of sulfuric acid
and water, which have refractive indices (around 1.45) more similar to DOS. The size distribution is presented here in 36 non-
uniformly spaced size bins.

The Meteorological Measurement Systems (MMS) provides high-resolution (20 Hz) measurements of ambient meteorological
parameters, including pressure, temperature, 3-dimensional wind vector, turbulence index, and GPS position, along the ER-2
flight track (Scott et al., 1990). The potential temperature coordinates are used to analyze the transport and evolution of plumes
in the stratosphere since stratospheric air masses tend to move along the lines of constant potential temperature (i.e.,
isentropes).



## 2.2 DCOTSS Balloon-borne Aerosol Measurements

Between 23[rd] and 28[th] of August 2021, five balloon soundings were carried out from Salina, Kansas (39° N, 98° W). In addition to a Particle Plus 8306 Optical Particle Counter (POPC) and a Compact Optical Backscatter Aerosol Detector (COBALD), the payloads also include a Cryogenic Frost-point Hygrometer (CFH) for water vapor mixing ratio measurements, and an iMET-

1 radiosonde for getting meteorological parameters and GPS locations. The POPC instrument (< 2 kg) was specifically developed for weather balloon deployments to make aerosol concentration profile measurement from the ground to the stratosphere. This profiling is achieved through the utilization of a 785 nm laser diode, capable of detecting particle diameters spanning eight bins ranging from 300 to 10,000 nm. A separate paper describing the system, its calibration procedure and comparisons with other instruments is in preparation. COBALD is a lightweight (540 g) instrument that consists of two high-

power light-emitting diodes (LEDs) that emit about 500 mW of optical power, at wavelengths of 470 and 940 nm, respectively. The backscattered light from the molecules, aerosols, or ice particles is recorded by a silicon photodiode using phase-sensitive detection. The scattering ratio calculated from COBALD measurements has an absolute error interval of 5%, with precision better than 1% in the upper troposphere and lower stratosphere (Vernier et al., 2015). The balloon flights reached altitudes of up to 30 km.

## 2.3 Satellite Observations

Two satellite instruments, the Infrared Atmospheric Sounding Interferometer (IASI) on board the MetOp-A, -B, and -C (MetOp) satellites and the TROPOspheric Monitoring Instrument (TROPOMI) on board the ESA/EU Copernicus Sentinel-5 Precursor (S5P) satellite were used to obtain the La Soufrière $SO_2$ vertical profile and burden for model input. IASI, a Fourier Transform spectrometer, exhibits good sensitivity to $SO_2$ within its spectral range (Scott et al., 1990). Using IASI, it is possible

to obtain $SO_2$ column amounts and heights for each pixel twice per day (Carboni et al., 2012). The column amounts and heights across the plume are aggregated to create a vertical distribution of $SO_2$ mass. Taylor et al. (2022) computed the vertical profile and burden of $SO_2$ emitted from the La Soufrière 2021 eruption using IASI/MetOp retrievals and found the bulk of the $SO_2$ plume to be near the tropopause region with approximately 31% in the stratosphere. Due to uncertainty in pixel height retrievals near the tropopause, the amount classified as stratospheric could be between 0.4% and 78%. The peak $SO_2$ burden derived

from IASI observations was $0.31\pm0.09$ Tg on April 13[th], which was lower than the 0.4 Tg peak burden derived from TROPOMI/S5P observations (Bruckert et al., 2023). As discussed by Taylor et al. (2022), IASI/MetOp $SO_2$ retrievals may be underestimated due to the effects of volcanic ash and the presence of $SO_2$ below the detection threshold of the instrument. Also, Vernier et al. (2023) pointed out the limited vertical sensitivity of current satellite sensors in $SO_2$ retrieval, which is around +/- 2 km in the lower stratosphere.

Following volcanic eruptions, the $SO_2$ injected into the stratosphere undergoes oxidation and transforms into sulfuric acid/water aerosol particles over a period of days to weeks. The Cloud-Aerosol Lidar with Orthogonal Polarization (CALIOP) instrument onboard the Cloud-Aerosol Lidar and Infrared Pathfinder Satellite Observations (CALIPSO) satellite provides



high-resolution vertical profiles of aerosols and clouds with near-global coverage. The CALIPSO satellite completes about 15 orbits per day with a 16-day repeat cycle, covering latitudes between 82° S and 82° N. The measurements of CALIOP are
based on the backscatter signals at 532 nm and 1064 nm wavelengths. Here we used the total attenuated backscatter at 532 nm from CALIOP Level 1 data version 4.11 following the data processing method developed by Vernier et al. (2009). Zonal averaging over 15 days was performed on the scattering ratios, calculated as the ratio of the measured total backscatter to the calculated molecular backscatter at 532 nm. The scattering ratio at 532 nm from CALIOP/CALIPSO provides an indicator of aerosol concentration.

### 2.4 SOCOL-AERv2 Model

SOCOL-AERv2 is a coupled aerosol-chemistry-climate model (Feinberg et al., 2019; Sheng et al., 2015). The chemistry-climate model SOCOL integrates the global circulation model MA-ECHAM5 with the chemistry model MEZON (Roeckner et al., 2003, 2004; Stenke et al., 2013). The third component, AER, is a sectional aerosol model, which describes sulfate aerosol microphysics and chemistry with interactive deposition schemes. The aerosols in SOCOL-AERv2 are divided into 40-size bins
of radii 0.39-3200 nm with the aerosol volume varying by a factor of two between each bin. The model resolution used in this study was T42 (2.8°x2.8°) with 39 vertical levels.

We prescribed the model simulation of the eruption using IASI/MetOp $SO_2$ plume retrievals reported by Taylor et al. (2022). We inputted the eruption over the entire day of April 12[th] using the IASI $SO_2$ vertical profile from April 12[th] (Table 1) to capture all stratospherically relevant eruptive events. For reasons discussed in Section 2.2, we scaled the IASI/MetOp $SO_2$
vertical profile to attain a total mass loading of 0.4 Tg of $SO_2$ as observed by TROPOMI/S5P on April 11[th] (Bruckert et al., 2023). The eruption was prescribed uniformly in the grid of 4-15° N, 20-62° W which corresponds to latitudes and longitudes where the fraction of the total $SO_2$ emission observed by IASI on April 12[th] was greater than 10% and 4%, respectively. To explore the sensitivity of our model simulations on the $SO_2$ vertical profile for reasons discussed in section 3.1, we ran two additional simulations: one with the IASI vertical profile shifted upward by 2 km and one with 0.1 Tg injected at 42 hPa (~22
km).

We performed the simulations with prescribed meteorology which nudges the model's temperature and wind fields to match ERA5 reanalysis (Hersbach et al., 2020). The model runs were initialized in 2015 with a constant carbonyl sulfide (OCS) surface mixing ratio of 500 ppt and $SO_2$ surface emissions based on RCP2.6 emissions projections for 2020 which results in a background stratospheric aerosol burden of 196 Gg S. We also performed controlled simulations without $SO_2$ injection to serve
as the non-eruption reference. To compare the model simulations with DPOPS measurements, we calculated the model number and surface area density using the aerosol size bins closest to the DPOPS measurement range, bins 24-36, which correspond to wet diameters of 158-2,536 nm.



## 3 Results and discussions

### 3.1 Distribution and transport of La Soufrière aerosol plumes

The distribution and transport of aerosols produced by the La Soufrière eruption are depicted in Figure 2, which shows CALIOP/CALIPSO 15-day zonal mean scattering ratios. The scattering ratio is an optical indicator of aerosol concentration but is also influenced by aerosol size. Figure 2a largely represents the background scattering ratio distribution before the La Soufrière eruption. In early May, approximately 3-5 weeks after the eruption, two distinct aerosol layers became apparent in the tropical stratosphere. The lower layer was centered around 18 km (~400 K), while the upper layer was centered around 21

km (~490 K). The lower plume dispersed poleward in the NH stratosphere along the isentropes and was within the DCOTSS measurement area (~30-55° N, ~8-21 km) during June, July, and August 2021. The lower plume was also gradually transported toward the southern hemisphere but at a relatively slower pace. The bulk of the upper plume did not display significant poleward transport but experienced dilution between April and September 2021. Routine balloon-borne measurements conducted over Boulder, Colorado (40° N, 105° W) in midlatitudes did not detect significant aerosol enhancement above 20

km throughout the year 2021 (Todt et al., 2023), indicating that the bulk of the upper plume did not reach the DCOTSS measurement area within NH midlatitudes.

Intriguingly, however, DCOTSS balloon-borne POPC measurements launched over Salina, Kansas detected a thin layer of aerosol enhancement for particles larger than 300 nm at 21.5 km on August 23, 2021 (Figure 3a). Subsequent balloon flights in late August, equipped with both POPC and COBALD instruments, also revealed an enhancement at the same altitude, albeit

with a reduced intensity (Figure S1 in the Supplement). This suggests the possibility of a transient filament excursion originating from the tropical upper plume. Notably, Figure 3a also displays a prominent enhancement below 20 km, attributed to the lower plume as observed by CALIOP observations.

Figure 3b shows that, for particles larger than 500 nm, the influence from upper plume becomes notably more pronounced, indicating the particle size difference between the upper and lower plumes. The upper plume likely consists of larger particles

due to an extended process time within the tropical reservoir, likely involving condensation growth. The variance in volcanic aerosol microphysical processes between the tropical reservoir and the midlatitude lower stratosphere, along with their consequent impact on changes in aerosol size, warrant further exploration.

Figure 4 shows the total aerosol surface area density (SAD) difference between injection and non-injection simulations performed by SOCOL-AERv2. The modeled and observed distribution and evolution of aerosol plumes agree well except our

model simulations do not reproduce the upper aerosol plume observed by CALIOP between 20-22 km in the tropics (Figure 2). This is because the IASI $SO_2$ vertical profile we used to prescribe the eruption does not indicate two distinct $SO_2$ plumes and suggests that 97% of the plume is below 20 km. IASI retrievals are more sensitive to where a large amount of $SO_2$ is and can potentially miss a smaller amount of $SO_2$ at higher altitudes when significant amounts of $SO_2$ are below it. By injecting 0.1 Tg of $SO_2$ in one grid box at 42 hPa (~22 km) in the model, we can reproduce an upper plume that, like in CALIOP





observations, remained relatively stagnant between May and September (Figure S4). The lack of a quantitative observational

constraint led us to not consider the upper plume in our simulations.

## 3.2 Performance of Model Simulations

As discussed in Section 3.1, the modeled plume dispersion agrees well with CALIOP/CALIPSO observations except for the

localized upper plume in the tropical region. Considering that the vertical sensitivity of current satellite $SO_2$ retrievals is around

+/- 2 km in the lower stratosphere (Vernier et al., 2023), an additional model run with the IASI $SO_2$ vertical profile shifted

upward by 2 km was conducted, which resulted in marginal changes to the aerosol SAD profiles in the NH midlatitudes. This

result, along with the lack of significant northern transport of the tropical upper plume to the DCOTSS sampling region within

NH midlatitudes, indicates that the discrepancy between model and CALIOP observations does not hinder our comparison

between model results and DCOTSS observations.

The SOCOL-AERv2 simulations of the La Soufrière eruption reproduced the volcanic perturbation with good agreement with

DCOTSS observations of SAD enhancement vertical profiles in the lower stratosphere (Figures 5c and 5d). However, the

model failed to reproduce the background aerosol conditions in the volcanically quiescent stratosphere. The modeled non-

injection SAD vertical profile is significantly elevated relative to DCOTSS 2022 observations (Figures 5a and 5b). This

discrepancy might arise from an overestimation of convective transport in the model, resulting in an excessive transportation

of aerosols and $SO_2$ to the upper troposphere (Feinberg et al., 2019). The elevated background in the model complicates the

direct comparison of model results to observations, particularly for the parameters where the volcanic perturbation cannot be

isolated, such as size distribution and effective diameter. The DCOTSS summer 2022 data is taken as a suitable baseline

reference in this study because (1) the data is comparable with the "unperturbed aerosol profile", which represents the median

of all profiles without discernible indications of recent volcanic or pyrogenic perturbation between March 2019 and March

2022, obtained from balloon-borne measurements conducted over Boulder, Colorado (Figure 3 in Todt et al., 2023), and (2)

SOCOL-AERv2 simulations indicate that aerosol concentration had declined close to non-injection values by June 2022 (See

Figures S2 and S3 in the Supplement).

## 3.3 Aerosol Concentration and Size Distribution from Aircraft Measurements

DCOTSS aircraft measurements found that the aerosol SAD in NH midlatitudes during summer 2021 was significantly

enhanced by a factor of 2-4 between 380-500 K relative to summer 2022 (Figure 5), which was minimally influenced by

volcanic activity. Figure S5 in the Supplement provides similar evidence of an enhancement in aerosol number concentration.

Over Palmdale, California (35° N, 118° W), the SAD profile peaked at around 400 K, with the SAD values ranging from 2 to

3.5 $\mu m^2/cm^3$ (Figure 5a). While over Salina, Kansas (39° N, 98° W), the SAD profile double-peaked at around 420 K and 450

K, with values ranging from 1.5 to 2.5 $\mu m^2/cm^3$ (Figure 5b). The observed differences in both the magnitude and shape of the

DCOTSS SAD profiles above Palmdale and Salina in 2021 suggest the presence of spatial variations in volcanic aerosols in





the stratosphere. Here we examine these variations in aerosol concentration and size distribution using detailed *in situ* aircraft measurements and aerosol-chemistry-climate model simulations.

Figure 6 presents the meridional variations in the particle number density profiles and effective diameter profiles, along with the corresponding size distribution at the peak of each profile. The effective diameter describes the area-weighted mean
diameter of the aerosol size distribution (Grainger, 2022). This parameter provides a simplified representation of the aerosol size distribution and is particularly relevant for characterizing aerosol-light interactions and heterogeneous chemistry.

Greater peak number densities were observed at high latitudes (Figure 6a). This aligns well with the transport of volcanic plumes shown in Figures 2 and 4, which indicate the bulk of the plume was already transported to high latitudes in July. The plume peak appears to decrease in potential temperature with northern transport which is likely the result of the combination
of poleward isentropic transport, which maintains a consistent potential temperature, and the downwelling Brewer-Dobson circulation in mid-high latitudes, responsible for descending the plume (Bönisch et al., 2011). The SOCOL-AERv2 simulations reproduced the influence of these large-scale air movements on the meridional variations of the particle number density profile (Figure S6a in the Supplement). Unlike the number density profiles, the effective diameter profiles and aerosol size distributions between 39° N and 54° N did not exhibit significant meridional variations (Figure 6b and 6c). Modeling results
of the effective diameter profiles and volume size distributions also support these findings (Figure S6b and S6c in the Supplement), though our model was unable to accurately reproduce the shape and magnitude of the observed effective diameter profile.

Similar to Figure 6, Figure 7 illustrates the zonal variations of particle number density profile, effective diameter profile, and the corresponding size distribution between 430 and 450 K potential temperature. In Figure 7a, distinct shapes of the number
density profile were observed between 85° W and 110° W, which were not replicated by the model simulation (see Figure S7a in the Supplement). This variation is likely due to dynamical variability associated with synoptic waves and wave breaking that is challenging to capture in the model (Charlton-Perez et al., 2013; Jing et al., 2020). Effective diameter profiles and aerosol size distributions between 85° W and 110° W exhibit minimal zonal variations, as depicted in Figure 7b and 7c. The modeled effective diameter and aerosol size distribution also exhibited minimal zonal variations (Figure S7b and S7c in the
Supplement).

Figures 6 and 7 collectively suggest that, at a given time and potential temperature level, there are minimal meridional and zonal variations in aerosol size within the stratospheric volcanic plumes; however, the aerosol effective diameter shows significant variability with altitude. Conversely, aerosol number concentration profiles within the volcanic plumes exhibit substantial meridional and zonal variations, primarily influenced by large-scale circulations and synoptic variability. For a
broader perspective, the modeled global depiction of spatial variations in volcanic aerosol concentration and their temporal evolutions is available in Figure S3 in the Supplement.

There is evidence that the La Soufrière eruption resulted in a smaller aerosol effective diameter in the midlatitude lower stratosphere. The effective diameter below 460 K over Salina (39° N, 98° W) exhibited smaller values in 2021 compared to 2022 (Figure 8a). Figure 8b further implies that the La Soufrière volcanic plumes contained a notably higher proportion of



small particles (< 400 nm) than the baseline condition observed in 2022, thereby contributing to the diminished aerosol
effective diameter over Salina in 2021. This finding is supported by routine balloon-borne measurements conducted over
Boulder, Colorado (40° N, 105° W), which show a consistent trend upon comparing data from the summers of 2021 and 2020,
as well as the spring of 2019 before the Raikoke eruption (see Figure S8 in the Supplement). The reduction in stratospheric
aerosol size following the 2021 La Soufrière eruption is also indicated by SAGE III/ISS satellite retrievals (Wrana et al., 2023).

Volcanic eruptions are generally thought to lead to an increase in aerosol size (Quaglia et al., 2023). However, the 2021 La
Soufrière eruption led to an increase in the number concentration of small particles (< 400 nm). The emergence of a notable
quantity of small particles is likely related to the latitude and altitude of the $SO_2$ injection, which in turn influences the aerosol
microphysical processes that regulate the aerosol size. For particles ranging from 100 and 600 nm, smaller particles are much
less efficient in scattering solar radiation (Murphy et al., 2021), resulting in a minor radiative impact of the 2021 La Soufrière

eruption, as shown in Table 2.

On the other hand, the effective diameter profile over Palmdale (35° N, 118° W) shows no noticeable difference between 2021
and 2022 (Figure 8c). It is worth noting that both the effective diameter profiles (2021 data in Figure 8a and 8c) and volume
size distributions (2021 data in Figure 8b and 8d) within the volcanic plumes exhibit similarities between Salina and Palmdale.
Consequently, the minimal variance in effective diameter observed over Palmdale between 2021 and 2022 can be attributed

to the already smaller effective diameters present in that region during the sampling period in 2022. The relatively short
DCOTSS measurement period (June 29-July 11) over Palmdale in 2022 might not sufficiently represent the averaged baseline
condition.

### 3.4 Ozone and Radiative Impacts

The La Soufrière eruption has been suggested to have potentially contributed to the exceptionally large Antarctic ozone hole

in 2021 (Yook et al., 2022). However, our model simulations show minor Arctic and Antarctic ozone loss with ozone column
depletion peaking at about 0.6% in 2021 (Figure 9), which is well within the natural variability. It is important to acknowledge
that the modeled ozone loss can be influenced by the specific $SO_2$ vertical profile used to prescribe the SOCOL-AERv2 model.
Given that our model simulations do not reproduce the upper aerosol plume observed by CALIOP/CALIPSO, the actual ozone
response might also potentially differ from our projected outcomes. However, the Antarctic ozone depletion in the simulation

with the IASI $SO_2$ vertical profile shifted higher by 2 km is less pronounced in 2021 (peak is ~ 0.3%) and more prominent in
2022 (peak is ~1.5%) (Figure S9a in the Supplement). Additionally, the simulation with 0.1 Tg of $SO_2$ injected at 22 km, which
reproduce the upper plume, only exhibits Antarctic ozone depletion in 2022 (peak is ~1%) (Figure S9b in the Supplement).
Although the two sensitivity runs yield different ozone column depletion compared to Figure 9, the values from all simulations
remain minimal and within the range of natural variability. Thus, we do not expect the La Soufrière eruption to have contributed

325   significantly to ozone loss in 2021. It is possible the 2019-2020 Australian wildfires, which caused a large ozone hole in 2020
(McNeill and Thornton, 2023; Rieger et al., 2021; Salawitch and McBride, 2022; Solomon et al., 2023), also contributed to



the similarly large Antarctic ozone hole of 2021 (https://ozonewatch.gsfc.nasa.gov/statistics/annual_data.html). The potentially long-lasting ozone loss caused by the Australian wildfires warrants further investigation.

Radiative forcing is calculated as the difference between the net radiative fluxes from the injection and no-injection model simulations. The modeled top-of-atmosphere one-year average global total radiative forcing was -0.08 W/m$^2$ clear-sky and -0.04 W/m$^2$ all-sky (Table 2). Table 2 also shows values of shortwave and longwave forcing, separately. The shortwave forcing from La Soufrière eruption is in the same range (no more than -0.05 W/m$^2$ all-sky) as the estimation for the 2019 Ulawun eruptions, which yielded a similar total SO$_2$ loading (~0.35 Tg). This shortwave forcing, however, is roughly one-third of that (-0.1 to -0.2 W/m$^2$ all-sky) estimated for the 2019 Raikoke eruption, which had a significantly larger total SO$_2$ loading (~1.35 Tg) (Kloss et al., 2021). Additionally, the one-year mean all-sky forcing from La Soufrière eruption represents about one-third of the -0.12 W/m$^2$ global multiannual mean volcanic forcing during the 2005-2015 period, marked by a series of small-to-moderate-magnitude explosive eruptions (Schmidt et al., 2018).

The clear-sky total radiative forcing was largest in June 2021 with a one-month average of -0.15 W/m$^2$ (see Figure S10 in the Supplement). The radiative effects were concentrated in the tropics and NH midlatitudes (0-70˚ N one-year average: -0.12 W/m$^2$ clear-sky and -0.07 W/m$^2$ all-sky) as the aerosol plume was transported minimally to the southern hemisphere (see Figure S3 in the Supplement for the global dispersion of the aerosol plume). The radiative effects diminished to near-baseline level (less than -0.03 W/m$^2$ clear-sky) after one year.

## 4 Conclusions

The eruption of La Soufrière in April 2021 resulted in two distinct enhanced aerosol layers in the tropical lower stratosphere. These layers emerged approximately 3-4 weeks after the eruption, specifically around 18 km (~400 K) and 21 km (~490 K), as observed through CALIOP/CALIPSO measurements. The SOCOL-AERv2 aerosol-chemistry-climate model reproduced the distribution and transport of the lower volcanic plume to higher latitudes in the NH, which was sampled extensively during the DCOTSS 2021 mission with the NASA ER-2 aircraft. DCOTSS measurements showed that aerosol SAD and number density in NH midlatitudes in the summer of 2021 were enhanced by a factor of 2-4 between 380-500 K potential temperature, relative to the minimally perturbed stratosphere at NH midlatitudes in the summer of 2022. Modeled aerosol enhancements in SAD and number density were consistent with the DCOTSS aircraft measurements. Although the tropical upper plume exhibited restricted poleward transport, it potentially influenced midlatitudes transiently, as indicated by DCOTSS balloon-borne measurements showcasing a transient aerosol layer around 21.5 km in late August 2021. These measurements implied that particles within the upper plume were larger than those present in the lower plume, likely due to an extended process time within the tropical reservoir.

This study examines the spatial variations in aerosol number concentration and size distribution of volcanic plumes using *in situ* aircraft measurements and model simulations. Both the aircraft measurements and model simulations revealed significant meridional variations in particle number density profiles in NH midlatitudes. These variations were attributed to the large-



scale movement of the volcanic plumes, resulting from a combination of poleward isentropic transport and downwelling Brewer-Dobson circulation in the midlatitudes. Aircraft measurements also show zonal variations of particle number density profile which were not replicated by the model, which may be the result of synoptic variability that is challenging to capture in global climate models. Notably, no significant meridional or zonal variations in the mode or shape of the size distribution were observed within the studied latitudes (39° N-54° N) and longitudes (89° W-110° W). The La Soufrière eruption resulted in a shift toward small particles during the summer of 2021, relative to the summer of 2022. However, no significant change in effective diameter was observed over Palmdale, possibly due to the already smaller values in that region during the limited sampling period in 2022. *In situ* aerosol concentration and size distribution measurements during DCOTSS mission conducted 2-4 months after the eruption, will be valuable for validating satellite retrievals and model simulations of stratospheric aerosols following small-moderate volcanic eruptions.

Our results suggest that the modeled ozone loss due to the 2021 La Soufrière eruption was minor, with a peak ozone column depletion of about 0.6% in both Arctic and Antarctic regions. This change falls within the natural variability and is unlikely to have contributed significantly to the exceptionally large Antarctic ozone hole in 2021. As a result, we speculate that the lingering effects of the Australian New Year wildfires in 2020 may have contributed significantly to the 2021 Antarctic ozone hole. The eruption's radiative impact was modeled to be -0.08 W/m$^2$ clear-sky and -0.04 W/m$^2$ all-sky. This all-sky forcing represents about one-third of the global multiannual mean volcanic forcing during the 2005-2015 moderate volcanic period. The radiative effects were concentrated in the tropics and NH midlatitudes and diminished to near-baseline level after one year.

**Data availability**

All aircraft and balloon-borne measurement data from the DCOTSS mission are publicly available at NASA Atmospheric Science Data Center: https://asdc.larc.nasa.gov/project/DCOTSS/DCOTSS-Aircraft-Data_1. CALIOP/CALIPSO satellite data are publicly available at: https://asdc.larc.nasa.gov/project/CALIPSO. Modeling data will be made available upon request.

**Author contributions**

YL, JD, JV, AKP, TB, and JDD collected and analyzed the *in situ* measurement data. CP performed the SOCOL-AERv2 model simulations with guidance from SV and AS. JV analyzed the CALIOP/CALIPSO satellite data. YL and CP wrote the manuscript with inputs from all the authors. FNK provided project guidance. All authors discussed, edited, and proofread the manuscript.



## Acknowledgments

This work was supported by the National Aeronautics and Space Administration under grant 80NSSC19K0326. The authors thank Gabriel Chiodo, Timofei Sukhodolov, and Thomas Peter from ETH Zürich for useful discussions on SOCOL-AERv2 simulations, Isabelle A. Taylor from the University of Oxford for providing IASI/MetOp $SO_2$ data, and Emrys Hall from

NOAA CSL for helpful discussions. The authors also acknowledge the whole DCOTSS team.

## Competing Interests

At least one of the authors is a member of the editorial board of *Atmospheric Chemistry and Physics*.

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



**Table 1. The IASI/MetOp SO₂ vertical profile on April 12ᵗʰ (Taylor et al., 2022) binned to the SOCOl-AERv2 model levels. The total SO₂ injected in the model is 0.4 Tg. The pressure altitudes of the model levels were calculated using the US Standard Atmosphere.**

| Model level (hPa) | Altitude (km) | SO₂ loading |
|---|---|---|
| 41.6 | 21.8 | 1 % |
| 54.0 | 20.1 | 2 % |
| 69.7 | 18.5 | 9 % |
| 89.1 | 16.9 | 49 % |
| 113.3 | 15.4 | 27 % |
| 142.9 | 13.9 | 6 % |
| 179.1 | 12.5 | 3 % |
| 223.0 | 11.1 | 2 % |
| 275.8 | 9.7 | 1 % |

**Table 2. Modeled radiative forcing from the La Soufrière eruption. The values are top-of-atmosphere one-year area-weighted global averages. Note that due to rounding errors, the sum of shortwave and longwave values may not precisely align with the total values.**

| | Shortwave | Longwave | Total |
|---|---|---|---|
| Clear-sky | -0.10 | +0.02 | -0.08 |
| All-sky | -0.05 | +0.01 | -0.04 |

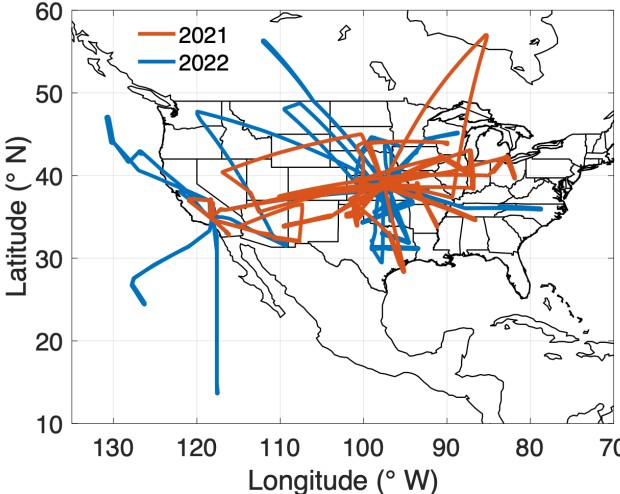

**Figure 1. Flight tracks (of 31 total flights) of the NASA ER-2 high-altitude research aircraft during the DCOTSS 2021 (June-August) and 2022 (May-July) missions.**





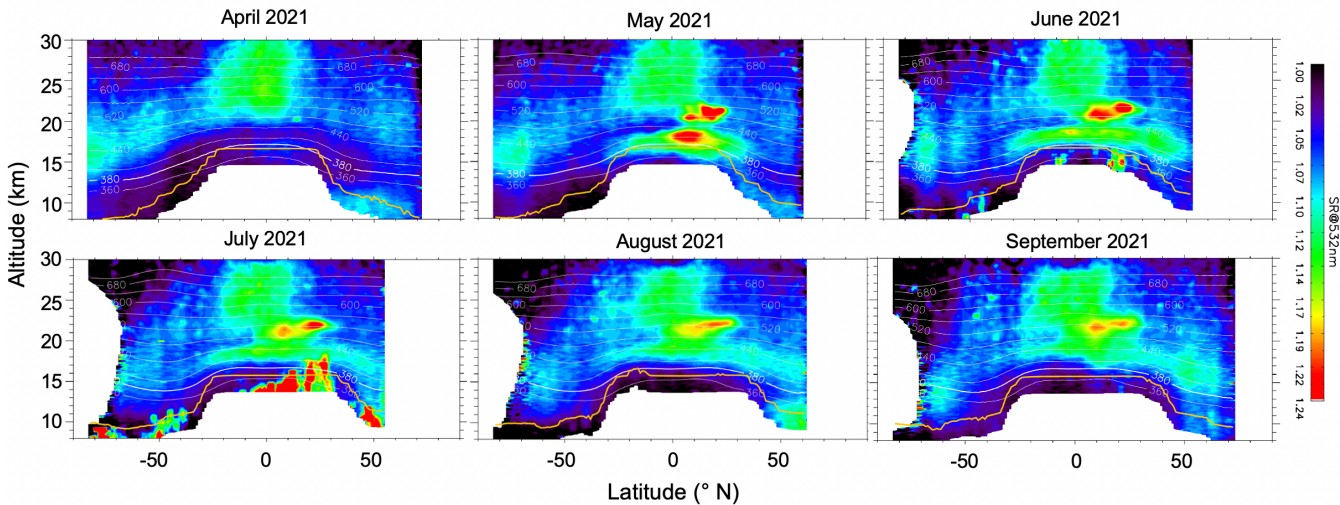

**Figure 2. Latitude and altitude distributions of scattering ratios (measured total backscatter divided by calculated molecular backscatter) at 532 nm from April to September 2021. Scatter ratio data are retrieved from CALIOP/CALIPSO observations and zonally averaged over the first 15 days of each month. Positive latitude values refer to the northern hemisphere and negative values refer to the southern hemisphere. The white lines and numbers indicate the potential temperature levels. The yellow lines indicate the tropopause.**

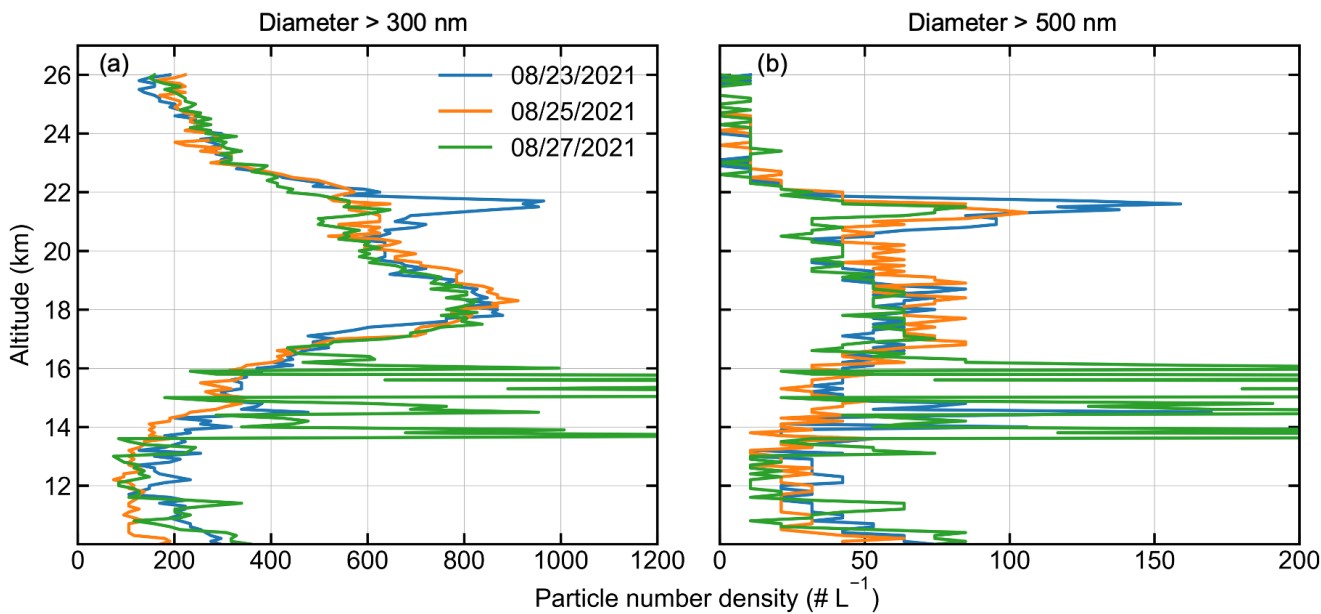

**Figure 3: Aerosol number density profiles from balloon-borne Particle Plus 8306 Optical Particle Counter (POPC) measurements for particles larger than 300 nm (panel a) and 500 nm (panel b) over Salina, Kansas.**



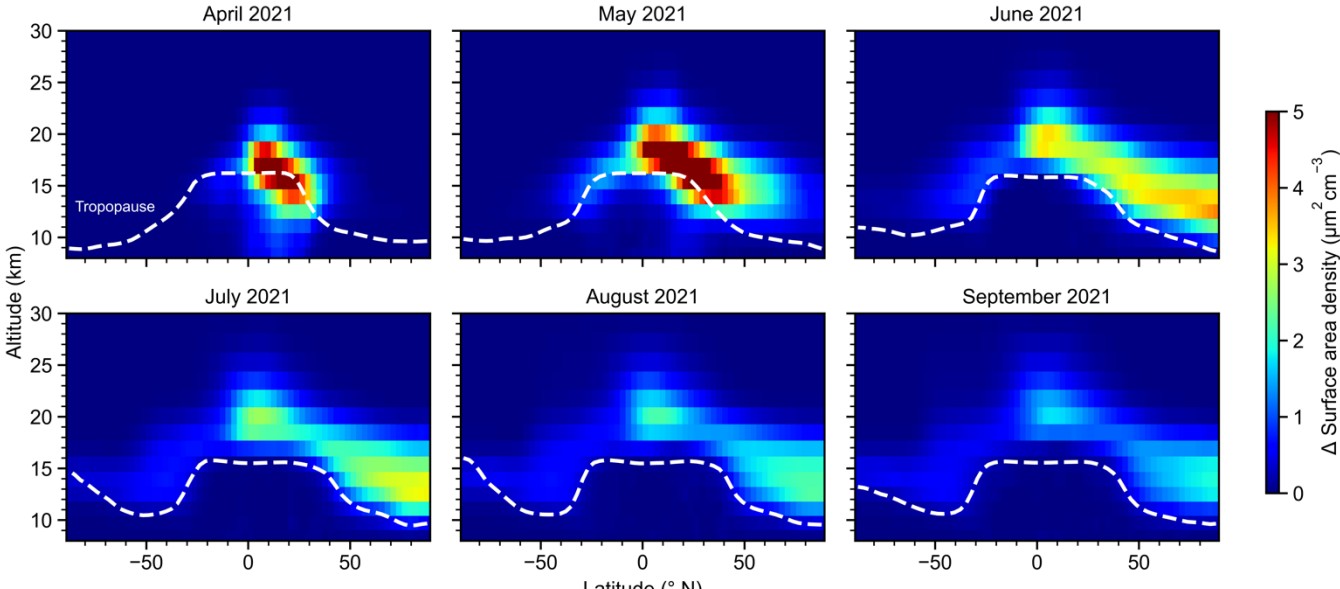

**Figure 4. Latitude and altitude distributions of aerosol surface area density (SAD) difference between injection and non-injection simulations performed by SOCOL-AERv2 from April to September 2021. Delta SAD was averaged zonally and monthly and used all aerosol size bins in the calculation. The white dashed lines indicate the WMO-defined tropopause.**

535



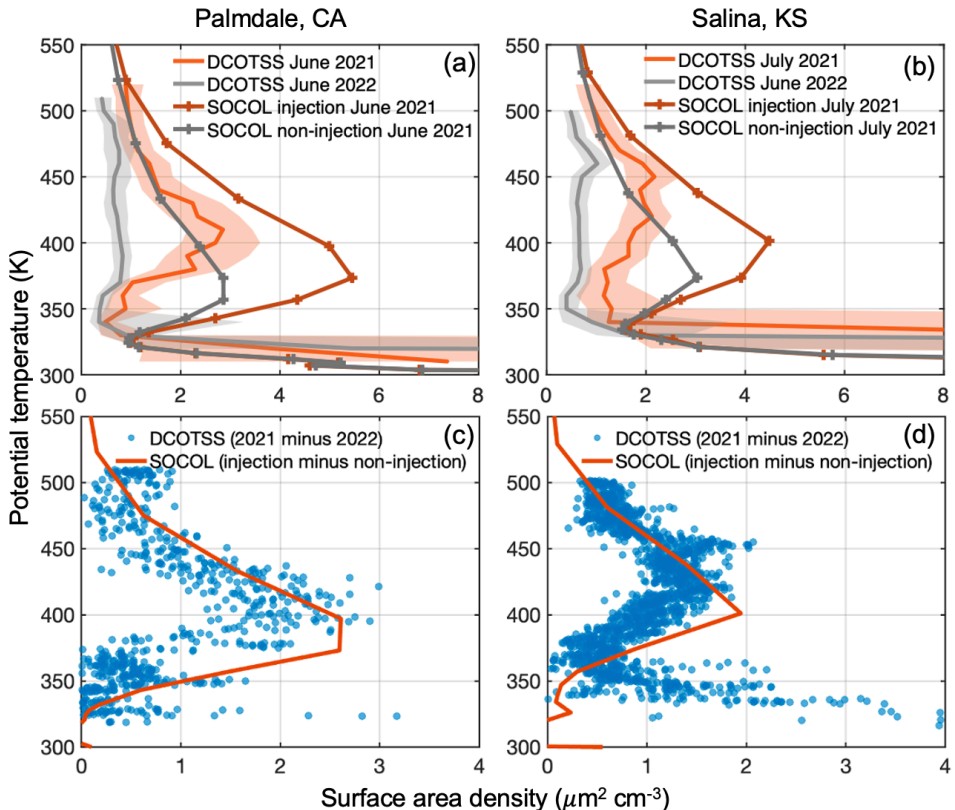

**Figure 5. Comparisons of aerosol surface area density (SAD) vertical profiles between SOCOL-AERv2 model simulations and DCOTSS aircraft measurements. (a) and (c) present vertical profiles over Palmdale (averaged over 32-38° N, 115-121° W), while (b) and (d) present vertical profiles over Salina (averaged over 36-42° N, 95-101° W). The absolute SAD profiles are presented in (a) and (b), while the SAD enhancement profiles are presented in (c) and (d). Shaded areas in (a) and (b) correspond the 10th and 90th percentiles of DCOTSS aircraft measurements.**



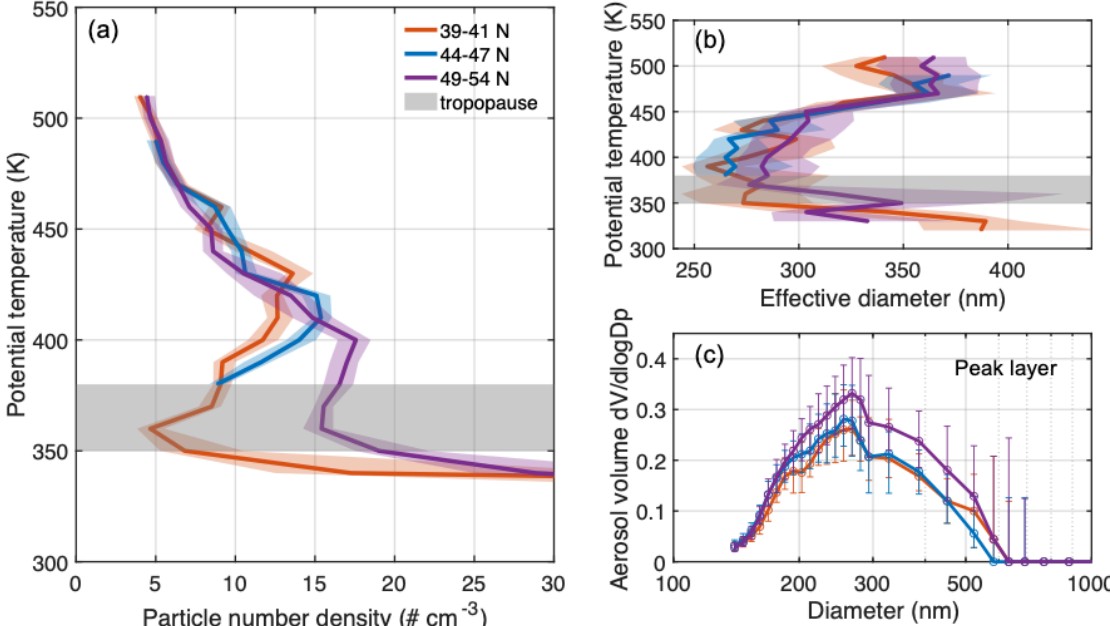

**Figure 6. Meridional variations of (a) aerosol number density profiles, (b) effective diameter profiles, and (c) aerosol volume density size distributions at the peak of each profile between 39° N and 54° N. All the data were collected during a single ER-2 flight on 16 July 2021, covering the region between 90° W and 100° W. Panel c presents aerosol size distribution at the peak of each profile shown in panel a. All lines represent the median values. Shaded areas in (a) and (b) and error bars in (c) represent the 25th and 75th percentiles.**





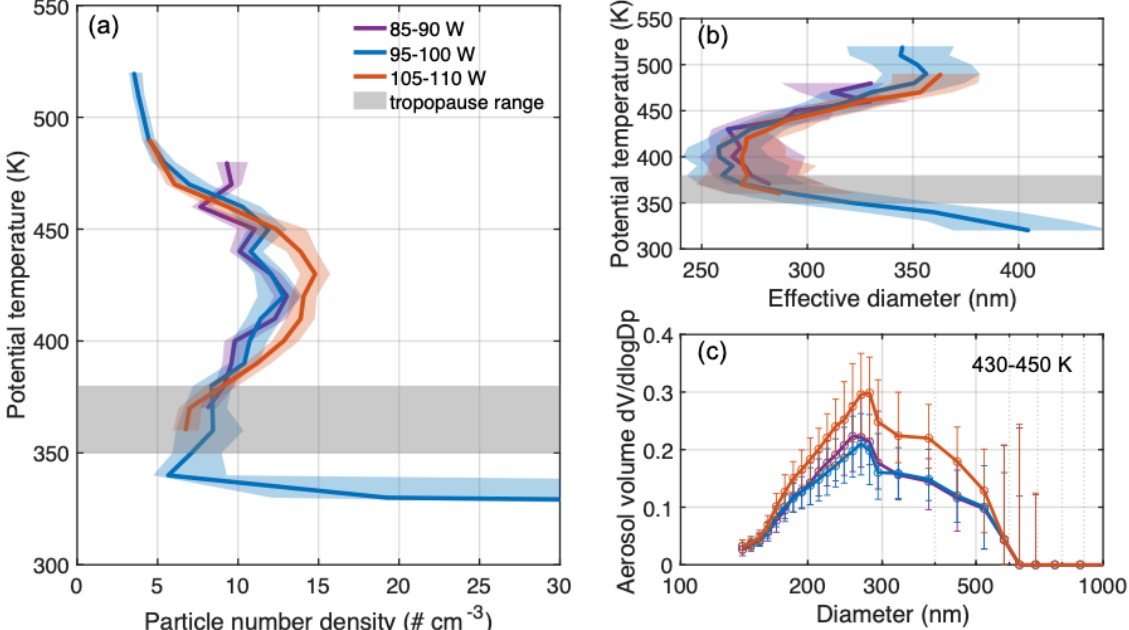

**Figure 7. Zonal variations of (a) aerosol number density profiles, (b) effective diameter profiles, and (c) aerosol volume density size distributions between 85° W and 110° W. All data were taken from 4 ER-2 flights between 20-29 July 2021 between 35° N and 40° N. Panel c presents aerosol size distributions between 430 and 450 K potential temperature range, where the most pronounced zonal variations in particle number density manifest. All lines represent median values. Shaded areas in (a) and (b) and error bars in (c) represent the 25th and 75th percentiles.**



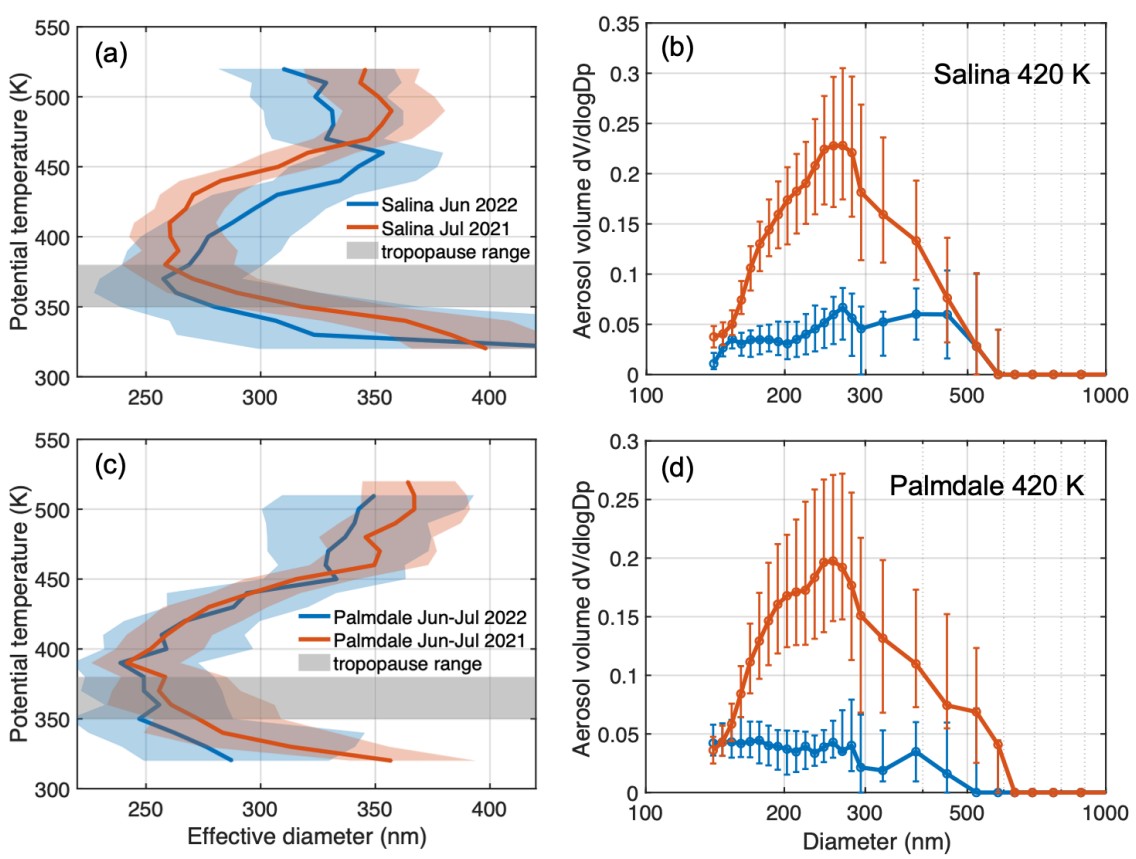

**Figure 8.** Aerosol effective diameter and volume density size distributions over Salina (averaged over 36-42° N, 95-101° W) and Palmdale (averaged over 32-38° N, 115-121° W) in 2021 and 2022 from DCOTSS aircraft measurements. Shaded areas and error bars represent the 25th and 75th percentiles.

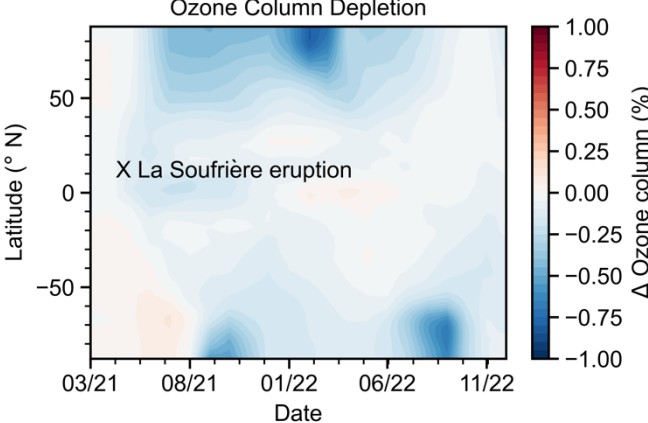

**Figure 9.** Zonal-averaged ozone column change following the La Soufrière eruption based on SOCOL-AERv2 simulations. Ozone column change is shown as the percent difference between the SO$_2$ injection simulation and the no-injection simulation.