# Peer review of "In situ measurements of perturbations to stratospheric aerosol and modeled ozone and radiative impacts following the 2021 La Soufrière eruption"

_EGUsphere, 2023_

## Referee Comment (RC2)

Review: In situ measurements of perturbations to stratospheric aerosol and modeled ozone and radiative impacts following the 2021 La Soufrière eruption

Li et al, 2023.

The paper addresses scientific questions that are well within the scope of ACP. The aerosol data collected in the DCOTSS stratospheric aerosol flights with the ER2 combined with balloon borne ascents in unique. The comparison between the SOCOL-AERv2 model and the observations is interesting.

The most prominent conclusion is that the aerosol size distribution in the lower stratosphere consists of more numerous smaller (presumably) sulfuric acid particles, with larger particles overlying them. Sound conclusions are drawn. The presentation in terms of clarity is good, with good figures and captions but a few clarifications and caveats are required. Some of the clarifications are important, and the authors should address them in a revision of the manuscript.

General comments:

1) Without a description of how ozone is impacted by aerosols within the SOCOL model, one cannot assess the model's treatment of any ozone depletion – at least a basic description of e.g. heterogeneous chemistry, treatment of PSCs should be given in section 2.4.
2) Any impacts on ozone from the eruption are likely to occur from a) heterogeneous chemistry, b) aerosol-induced stratospheric heating which can change the poleward transport of ozone. Given that you are using nudged simulations, the dynamically induced response is likely to be suppressed. This caveat should be included.
3) In the modelling, some mention of the approximate number of model layers in the stratosphere should be mentioned together with the model top. From looking at the supplement (e.g. S5) it appears that the resolution in the stratosphere is quite limited with 5-6 layers being represented.
4) Some acknowledgment of the limitations of model resolution (spatial and temporal) should be made. For example, the study of the combined eruption of Raikoke using a global model with a resolution of around 10km and 59model levels (e.g. de Leuuw et al., 2021, https://doi.org/10.5194/acp-21-10851-2021; Osborne et al., 2022; https://doi.org/10.5194/acp-22-2975-2022) does not have to make an injection into such a large area. The detailed evolution and 'filamentary' structure that is referred to in the text in this study is difficult to model with such a crude injection strategy and such a coarse resolution and an appropriate caveat should be made.

[Figure]

Left panel – SO2 (de Leuuw et al., 2021); right panel sulfate and ash (Osborne et al., 2022).

5) There needs to be an acknowledgement that comparison between Fig 2 and Fig 4 is not a like-like comparison as Fig 2 includes the background aerosol while Figure 4 does not. This is mentioned later in the paper, but needs to be acknowledged sooner.

6) In section 3.2 the statement, "As discussed in Section 3.1, the modelled plume agrees well with CALIOP/CALIPSO ……" is rather pushing it. There is one sentence in section 3.1, which does not constitute a discussion. There is no quantitative analysis that supports the 'good agreement' – its just done by eyeballing the two plots. Ideally the background could be removed from multi-year CALIPSO data. Then you'd be much closer to a like-like comparison and could provide quantitative numbers to back up your text.

7) Do the authors think that the rapid change in the observed size distribution with diameter by the POPS (which is a nice bit of lightweight kit) at 300nm diameter is real? I know that there have been some comparisons between the POPS and other instrumentation such as the SMPS which operate quite differently in terms of physical measurements, and there seem to be some differences in the slope of the size distribution that is derived between the two instruments (e.g. Liu et al., 2021; https://doi.org/10.5194/amt-14-6101-2021). There has been some discussion with Handix as to whether the cross over in gain stages of the pre-amplifier/amplifier could have some influence. Whatever, the case, it does seem a notable feature throughout the results that are presented here.

Specific comments:

8) L121 – size distribution – is this radius or diameter? Actually you can find that this is diameter (L198), but diameter should be stated here.

9) The size distributions and effective diameters in Figure S6 & S7 should be the same as those in Figure 6 and 7 to aid comparison.

10) The white lines and numbers in Fig 2 should be boldened.

---

## Author Comment (AC1)

**Response to reviews**

We thank the reviewers for their constructive comments to improve the manuscript. Below we provide point-by-point responses to these comments, and make changes to the manuscript as necessary. Reviewer comments are in **bold**. Author responses are in plain text labeled with [R]. Line numbers in the responses correspond to those in the clean version of the revised manuscript. Modifications to the manuscript are in *italics*.

**RC1 (Reviewer #1, Daniele Visioni)**

**This paper is a really great work that looks at how the eruption of La Soufrière affected the stratospheric aerosol layer through a mix of observations and modeling, which is always a great approach! The paper is very well done, comprehensive and well written while staying terse, so I don't have much to add, and I fully endorse publication in ACP. Just a few comments below.**

[R0] We thank the reviewer for the valuable feedback and constructive suggestions. Detailed responses are given below.

**L 43: I would add a "Additionally, we also modeled the eruption with…" to make it sound more like the two things are connected and follow from one another!**

[R1] We thank the reviewer for the suggestion. We have changed the sentence to reflect this comment, see Line 43.

**L 300: I think this point could be expressed better: it's sort of misleading as of now. Quaglia et al. (2023) was about Pinatubo, which is very large, and we have numerous measurements demonstrating the increase in size distribution. This paper and Wrana et al. (2023) both point out that smaller eruptions could decrease size actually, which is interesting! But also a different ballpark compared to Pinatubo… So keep the Quaglia et al. (2023) reference, but be clearer in specifying that those are two entirely different observations and not in conflict with each other.**

[R2] We thank the reviewer for clarifying this. We have added a few sentences in Line 312-315 to make it clearer: "*Volcanic eruptions have generally been considered to lead to an increase in aerosol size, as was clearly the case following massive events like the 1991 Pinatubo eruption (Quaglia et al., 2023). However, the present results show that this is not necessarily the case for all eruptions, especially for relatively smaller ones like the 2021 La Soufrière eruption, which are more frequent. La Soufrière eruption led to an increase in the number concentration of small particles (< 400 nm).*"

**L 329: specify this is ERF and not IRF, as (even though it's not exactly specified) SOCOL is run here with fixed SSTs.**

[R3] We thank the reviewer for pointing out this point of confusion. In this paper, RF is calculated as the difference in net TOA radiative fluxes between the injection and the non-injection scenario. Both the injection and non-injection simulations are nudged to ERA5 reanalysis of temperature and wind fields. Because of the nudging for both cases, the effects of atmospheric temperature and circulation adjustments are suppressed.

Furthermore, SOCOL-AERv2 does not include aerosol-cloud interactions. Thus, based on the definition of ERF put forth by Myhre et al., 2013 ("change in the net TOA downward radiative flux after allowing for atmospheric temperatures, water vapor and clouds to adjust, but with surface temperature or a portion of surface conditions unchanged."), we do not think that ERF is the appropriate designation.

We also agree with the reviewer that it is not IRF because the chemical response to the eruption (water vapor, ozone, etc.) is not suppressed. Additionally, the meteorological response to the eruption is relaxed – not fixed – to ERA5 reanalysis.

Thus, our RF calculation is neither ERF nor IRF. To clarify the nuance of our calculation, we added the following text to the manuscript (Line 351-353): *"Radiative forcing is calculated as the difference between the net top-of-atmosphere radiative fluxes from the injection and no-injection model simulations. It is important to note that, as discussed in the previous paragraph, the modeled dynamical response to the eruption is suppressed due to nudging to ERA5 reanalysis in both the injection and non-injection simulations."* (Note: the previous paragraph has been updated to discuss the suppressed dynamical response).

**L 372: a bit uncomfortable with the speculation about the fires here. Not the point of the paper, you didn't check for it, so sounds a bit unsubstantiated.**

[R4] We agree with the reviewer that this speculative statement is not appropriate for inclusion in the Conclusion section. Consequently, we have removed this sentence.

**RC2 (Reviewer #2, J.M. Haywood)**

**The paper addresses scientific questions that are well within the scope of ACP. The aerosol data collected in the DCOTSS stratospheric aerosol flights with the ER2 combined with balloon borne ascents in unique. The comparison between the SOCOL-AERv2 model and the observations is interesting.**

**The most prominent conclusion is that the aerosol size distribution in the lower stratosphere consists of more numerous smaller (presumably) sulfuric acid particles, with larger particles overlying them. Sound conclusions are drawn. The presentation in terms of clarity is good, with good figures and captions but a few clarifications and caveats are required. Some of the clarifications are important, and the authors should address them in a revision of the manuscript.**

[R0] We thank the reviewer for the valuable feedback and constructive suggestions. Detailed responses are given below.

**General comments:**

**Without a description of how ozone is impacted by aerosols within the SOCOL model, one cannot assess the model's treatment of any ozone depletion – at least a basic description of e.g. heterogeneous chemistry, treatment of PSCs should be given in section 2.4.**

[R1] We appreciate the reviewer for pointing out the lack of clarity in the model's treatment of ozone-relevant chemistry. A sentence has been added to briefly summarize the scope of the chemistry in the model and to point out Stenke et al., 2013 which has a detailed description of the chemistry, see Line 181-183: "*The model calculates a variety of chemical reactions including photolysis reactions, gas phase reactions, and heterogeneous reactions on sulfate aerosol and three types of polar stratospheric clouds (see Stenke et al., 2013 and references therein for a complete description of the chemistry).*"

**Any impacts on ozone from the eruption are likely to occur from a) heterogeneous chemistry, b) aerosol-induced stratospheric heating which can change the poleward transport of ozone. Given that you are using nudged simulations, the dynamically induced response is likely to be suppressed. This caveat should be included.**

[R2] We thank the reviewer for pointing out this important caveat. We have added sentences in Section 3.4 to acknowledge this, see Line 334-338: "*An important caveat is that both the injection and no-injection simulations are nudged toward ERA5 reanalysis which, to some degree, includes the dynamical response to the eruption. Thus, in our perturbation calculations (injection minus no-injection), the dynamical response captured by ERA5 reanalysis is partially subtracted out. As a result, the modeled ozone response is primarily due to heterogeneous chemistry.*"

**In the modelling, some mention of the approximate number of model layers in the stratosphere should be mentioned together with the model top. From looking at the supplement (e.g. S5) it appears that the resolution in the stratosphere is quite limited with 5-6 layers being represented.**

[R3] We thank the reviewer for the suggestion. Text has been added to Line 186-187 that includes the model top and base along with the range of model levels in the stratosphere: "*The model resolution used in this study was T42 (2.8˚x2.8˚) with 39 vertical levels from 1013.25 to 0.01 hPa. The stratosphere is represented by 15-20 model levels.*"

It is important to note that Figure S5 in the supplement only shows part of the stratosphere (up to 550 K potential temperature), the whole stratosphere in the model is represented by 15-20 model levels.

**Some acknowledgment of the limitations of model resolution (spatial and temporal) should be made. For example, the study of the combined eruption of Raikoke using a global model with a resolution of around 10km and 59 model levels (e.g. de Leuuw et al., 2021, https://doi.org/10.5194/acp-21-10851-2021; Osborne et al., 2022; https://doi.org/10.5194/acp-22-2975-2022) does not have to make an injection into such a large area. The detailed evolution and 'filamentary' structure that is referred to in the text in this study is difficult to model with such a crude injection strategy and such a coarse resolution and an appropriate caveat should be made.**

[R4] We agree with the reviewer that the model's spatial resolution is too coarse to resolve the fine-scale filamentary structure of the plume that was observed. The purpose of our model simulations was to investigate large-scale phenomena that are appropriate for the scale of the model.

The determination of the extensive area of SO$_2$ injection was based on the IASI satellite observations. On April 12[th], the day we prescribe the eruption in our model, IASI SO$_2$ column retrievals (Figure 7; Taylor et al., 2022; https://acp.copernicus.org/preprints/acp-2022-772/) display minimal filamentary structure. As we described in the original manuscript, SO$_2$ was prescribed in the "*grid of 4-15˚N, 20-62˚W which corresponds to latitudes and longitudes where the fraction of the total SO$_2$ emission observed by IASI on April 12[th] was greater than 10% and 4%, respectively*". That being said, the uniform injection scenario could potentially smooth out the fine structures; though we speculate that the inability of the model to reproduce the filamentary structure is primarily due to the coarseness of the model resolution, with the injection strategy playing a secondary role. To acknowledge this limitation, we have added a sentence in Line 193-196: "*Given the limited spatial resolution of our model (2.8˚x2.8˚ horizontal resolution, 39 vertical levels) and the adoption of a uniform injection scenario, simulating the fine-scale structure of volcanic plumes is not possible. Our primary focus is on large-scale phenomena that are appropriate for the scale of the model.*"

**There needs to be an acknowledgement that comparison between Fig 2 and Fig 4 is not a like-like comparison as Fig 2 includes the background aerosol while Figure 4 does not. This is mentioned later in the paper, but needs to be acknowledged sooner.**

[R5] We agree with the reviewer that the comparison between Fig 2 and Fig 4 is not a direct like-like comparison. In response to this, we have included an acknowledgement of this point in Line 232-235: "*Although Figure 4 and Figure 2 present different metrics (SAD vs. scattering ratio) for stratospheric aerosols, and Figure 2 incorporates the*

*background aerosols while Figure 4 does not, it is evident that there is a qualitative agreement…*"

**In section 3.2 the statement, "As discussed in Section 3.1, the modelled plume agrees well with CALIOP/CALIPSO ……" is rather pushing it. There is one sentence in section 3.1, which does not constitute a discussion. There is no quantitative analysis that supports the 'good agreement' – its just done by eyeballing the two plots. Ideally the background could be removed from multi-year CALIPSO data. Then you'd be much closer to a like-like comparison and could provide quantitative numbers to back up your text.**

[R6] We appreciate the reviewer for pointing out the lack of clarity. As addressed in response to the last comment ([R5]), we have included sentences in Section 3.1 to acknowledge the qualitative nature of the comparison between our model simulations (Figure 4) and CALIOP observations (Figure 2). Additionally, we have made it clearer that the qualitative agreement extends to the "*altitude, meridional distribution, and temporal evolution of aerosol plumes*" in Line 234-235.

We agree that removing the background from CALIPSO data in Figure 2 could facilitate a more comparable evaluation with Figure 4. However, it's worth noting that the background stratospheric aerosol layer is very often spatially modulated by the QBO, making the determination of a true background value a challenging task. Even if we manage to address this for Figure 2, there remains the distinction between the metrics used, as Figure 2 relies on scattering ratio, while Figure 4 utilizes aerosol surface area density (SAD). In response to this, we have refined the statement in Section 3.2 to be: "*As discussed in Section 3.1, the modeled plume dispersion agrees qualitatively with CALIOP/CALIPSO observations…*", see Line 244.

We believe that conducting a qualitative comparison between model simulations and CALIPSO observations is adequate, given that Section 3.2 encompasses more quantitative assessments of modelled SAD enhancements through direct comparisons with DCOTSS *in situ* measurements.

**Do the authors think that the rapid change in the observed size distribution with diameter by the POPS (which is a nice bit of lightweight kit) at 300nm diameter is real? I know that there have been some comparisons between the POPS and other instrumentation such as the SMPS which operate quite differently in terms of physical measurements, and there seem to be some differences in the slope of the size distribution that is derived between the two instruments (e.g. Liu et al., 2021; https://doi.org/10.5194/amt-14-6101-2021). There has been some discussion with Handix as to whether the cross over in gain stages of the pre-amplifier/amplifier could have some influence. Whatever, the case, it does seem a notable feature throughout the results that are presented here.**

[R7] We appreciate the reviewer's meticulous observation and valuable insights into the technical aspects of the POPS instrument. After careful investigations, we believe the rapid change around 300 nm diameter is an artifact resulting from the use of narrow size bins (specifically, 4 bins between 261 and 355 nm) in a region where the Mie curve is relatively flat, as illustrated below. In response to this concern, we have addressed the issue by combining these 4 bins into a single bin and subsequently revised Figures 6, 7, and 8 with the updated bin structure. As a result of these adjustments, the artifact

has been effectively eliminated.

[Figure]

We also conducted comparisons of both the counting efficiency and sizing accuracy of the DPOPS instrument against the SMPS instrument, to rule out any potential instrument detection or hardware-related issues. The plot on the left illustrates our counting efficiency calibration, showing that the counting efficiency for 300 nm DOS particles is 100±8%. On the right, the plot displays sizing accuracy for 300 nm DOS particles selected by DMA, with the detected mode at 294 nm and a reasonable distribution. In light of these, we have confidence in the DPOPS instrument in accurately counting and sizing particles around 300 nm with a suitable binning scheme. A separate paper providing detailed characterizations of the DPOPS instrument is currently in preparation.

[Figure]

**Specific comments:**

**L121 – size distribution – is this radius or diameter? Actually you can find that this is diameter (L198), but diameter should be stated here.**

[R8] Thank you for the point of clarification. We have added "*diameter*" to the sentence in reference, see Line 122.

**The size distributions and effective diameters in Figure S6 & S7 should be the same as those in Figure 6 and 7 to aid comparison.**

[R9] Thank you for the suggestion. We have updated Figures 6, 7, S6, and S7 to ensure consistent diameter ranges.

**The white lines and numbers in Fig 2 should be boldened.**

[R10] We have updated the Figure 2 with the white lines boldened.

**RC3 (Reviewer #3, Anonymous Referee)**

**120-150 It might be useful to include flow rates of the aerosol instruments and estimates of their minimum concentration sensitivity.**

[R1] We thank the reviewer for the suggestion. We have incorporated additional information into Section 2.1 to detail the flow rate and the estimated particle concentration range for the DPOPS instrument. This information can be found in Line 126-128: "*The sample flow rate is 0.8 LPM (liter per minute). At this flow rate, the DPOPS instrument is capable of measuring total particle number concentrations ranging from 0 to 150 #/cm³ with a coincidence error of less than 10%.*" We have also included the flow rate for the balloon-borne POPC instrument in Section 2.2, see Line 146: "*The sample flow rate for the POPC instrument is 2.83 LPM.*" A more comprehensive characterization of the balloon-borne POPC instrument is currently in progress and is intended for a separate paper.

**289 Are there really substantial zonal variations in number density. The profiles in Fig. 7a have the same shape and for 86-100 W the same values, with the profiles 105-110 W being only slightly higher.**

[R2] We appreciate the reviewer for the meticulous examination of the data. Indeed, zonal variations are less pronounced than meridional variations, especially for profiles within 85-100 W, where the shape remains similar. However, it's important to note that the profile within 105-110 W exhibits a distinct shape compared to those within 85-100 W.

To further illustrate this, we have conducted a quantitative analysis by calculating the percentage difference in particle number density profiles between 95-100 W and 105-110 W in comparison to the 85-90 W profile. The plot provided below demonstrates that variations in values are still considerable, with a maximum percentage difference of approximately +/- 40%.

In light of these findings, we have refined our statement in Line 300 to read: "*aerosol number concentration profiles within the volcanic plumes exhibit substantial meridional variations and, to a somewhat lesser extent, zonal variations.*"

[Figure]

**293-294 This is only true above the tropopause region.**

[R3] We have clarified this in Line 304: "*The effective diameter between the tropopause and 460 K over Salina (39° N, 98° W) exhibited smaller values in 2021 compared to 2022 (Figure 8a).*"

**Fig. S1. With a few assumptions about index of refraction the POPC aerosol size distribution measurements could be converted to a backscatter for direct comparison with COBALD. That would be a nice addition to more realistically compare these two independent measurements, than a simple profile of backscatter and number concentration at one channel.**

[R4] We appreciate the reviewer's suggestion, but due to the limited number of channels (8 channels) in our current POPC version, converting POPC data to an equivalent scattering ratio would introduce significant uncertainty. Figure S1 in the Supplement is intended to showcase two independent measurements of aerosol layers obtained from POPC and COBALD, not a direct comparison between these two instruments. Another paper is in preparation where we will address this by comparing COBALD with a new 30-channel version of POPC.

**310-312 It would be good to include on the figures the date span of the measurements, e.g. for Palmdale June 29 – July 11. This is hardly June-July, as presently noted on the figure. The same could be done for the Salina site.**

[R5] We thank the review for the suggestion. We have included date span of the measurements for both the Salina and Palmdale plots in Figure 8.

**314-328 Given the clearly demonstrated impact of La Soufriere on northern hemisphere aerosol and minimal impact in the southern hemisphere, Figs. 2 and 4, does it make sense to discuss Antarctic ozone loss due to La Soufriere? What is the mechanism by which this would occur? What is the argument of Yook et al. [2022] to show an impact on southern hemisphere ozone? This all seems a big stretch.**

[R6] We thank the reviewer for pointing out this confusion. Indeed, the aerosol impacts of the La Soufrière eruption were primarily concentrated in the northern hemisphere, while some volcanic plumes also reached the southern hemisphere, leading to a slight but noticeable increase in aerosol loading in the southern hemisphere. This clarification has been incorporated at the beginning of Section 3.4: "*As depicted in Figures 2, 4, and S3 in the Supplement, aerosol enhancements resulting from the La Soufrière eruption were predominantly concentrated in the northern hemisphere, although aerosol plumes were also transported to the southern hemisphere. Figure 9 shows the simulated column ozone loss induced by the La Soufrière eruption. While the signal was more pronounced in the northern hemisphere than in the southern hemisphere, the overall magnitude of ozone loss was relatively minor.*", see Line 328-332.

The rationale for discussing Antarctic ozone loss is grounded in previous studies, such as Yook et al. (2022), which documented the significant Antarctic ozone hole in 2021. Yook et al. (2022) speculated that the 2021 Antarctic ozone hole might be attributed to the La Soufrière eruption and/or the 2020 Australian wildfires. It's important to note that they did not establish a direct causal link between the La Soufrière eruption and

the 2021 Antarctic ozone hole. Rather, their analysis was speculative, exploring correlations between enhanced aerosol loading and ozone depletion in the southern hemisphere after the La Soufrière eruption.

Our study, on the other hand, serves to provide empirical evidence that the 2021 Antarctic ozone hole was not a result of the La Soufrière eruption. We believe this clarification is important to guide future research concerning the potential drivers of the 2021 Antarctic ozone hole.

**RC4 (Reviewer #3, Anonymous Referee)**

**I forgot in my review to complement the authors on a clear well written paper.**

[R1] We are grateful for the reviewer's kind words and extend our thanks for the valuable feedback and constructive suggestions provided.